# Leptin Levels of the Perinatal Period Shape Offspring’s Weight Trajectories through the First Year of Age

**DOI:** 10.3390/nu14071451

**Published:** 2022-03-30

**Authors:** Francesca Garofoli, Iolanda Mazzucchelli, Micol Angelini, Catherine Klersy, Virginia Valeria Ferretti, Barbara Gardella, Giulia Vittoria Carletti, Arsenio Spinillo, Chryssoula Tzialla, Stefano Ghirardello

**Affiliations:** 1Neonatal Unit and Neonatal Intensive Care Unit, Fondazione IRCCS Policlinico San Matteo, 27100 Pavia, Italy; m.angelini@smatteo.pv.it (M.A.); chryssoula.tzialla@unipv.it (C.T.); s.ghirardello@smatteo.pv.it (S.G.); 2Unit of Rheumatology, Department of Internal Medicine and Therapeutics, Università di Pavia, 27100 Pavia, Italy; iolanda.mazzucchelli@unipv.it; 3Clinical Epidemiology and Biostatistics Unit, Fondazione IRCCS Policlinico San Matteo, 27100 Pavia, Italy; klersy@smatteo.pv.it (C.K.); v.ferretti@smatteo.pv.it (V.V.F.); 4Unit of Obstetrics and Gynaecology, Fondazione IRCCS Policlinico San Matteo, 27100 Pavia, Italy; b.gardella@smatteo.pv.it (B.G.); g.carletti@smatteo.pv.it (G.V.C.); a.spinillo@smatteo.pv.it (A.S.)

**Keywords:** leptin, preterm neonate, intrauterine growth restriction (IUGR), growth trajectories, children health prognosis

## Abstract

Background: Leptin is a hormone regulating lifetime energy homeostasis and metabolism and its concentration is important starting from prenatal life. We aimed to investigate the association of perinatal leptin concentrations with growth trajectories during the first year of life. Methods: Prospective, longitudinal study, measuring leptin concentration in maternal plasma before delivery, cord blood (CB), and mature breast milk and correlating their impact on neonate’s bodyweight from birth to 1 year of age, in 16 full-term (FT), 16 preterm (PT), and 13 intrauterine growth-restricted (IUGR) neonates. Results: Maternal leptin concentrations were highest in the PT group, followed by IUGR and FT, with no statistical differences among groups (*p* = 0.213). CB leptin concentrations were significantly higher in FT compared with PT and IUGR neonates (PT vs. FT; IUGR vs. FT: *p* < 0.001). Maternal milk leptin concentrations were low, with no difference among groups. Maternal leptin and milk concentrations were negatively associated with all the neonates’ weight changes (*p* = 0.017 and *p* = 0.006), while the association with CB leptin was not significant (*p* = 0.051). Considering each subgroup individually, statistical analysis confirmed the previous results in PT and IUGR infants, with the highest value in the PT subgroup. In addition, this group’s results negatively correlated with CB leptin (*p* = 0.026) and showed the largest % weight increase. Conclusions: Leptin might play a role in neonatal growth trajectories, characterized by an inverse correlation with maternal plasma and milk. PT infants showed the highest correlation with hormone levels, regardless of source, seeming the most affected group by leptin guidance. Low leptin levels appeared to contribute to critical neonates’ ability to recover a correct body weight at 1 year. An eventual non-physiological “catch-up growth” should be monitored, and leptin perinatal levels may be an indicative tool. Further investigations are needed to strengthen the results.

## 1. Introduction

The adipose tissue is a dynamic endocrine organ that produces and secretes various adipokines and bioactive peptides, including leptin. Leptin is a hormone (16KD) synthesized by the obese (*ob*) gene, [1] the secreted amount is directly related to the mass of adipocytes. Leptin activates anorexigenic neuropeptide circuits and inhibits orexigenic routes after binding at hypothalamic receptors, decreasing food intake, and increasing energy expenditure.

Leptin stimulates lipid catabolism while constraining lipogenesis at the adipose cells level, through both central and peripheral energy homeostasis control and metabolism balance [2]. In addition, the physiological leptin concentration indicates that the body has sufficient energy stores, thus inhibiting appetite [3].

The role of leptin in the overall metabolism regulation starting at prenatal life has been previously suggested. In particular, the interplay between maternal and fetal leptin concentration might anticipate the aptitude to develop metabolic diseases in adulthood. Furthermore, maternal plasma, cord blood, and breast milk leptin concentrations from birth and during the first years of life seem good predictors of growth outline later in life [4,5,6,7,8,9,10,11,12,13].

Maternal leptin secretion rises during gestation due to the growing fetus and fat and body fluid increase. Maternal excessive weight gain during gestation is associated with gestational diabetes, pre-eclampsia, polycystic ovary syndrome, and preterm or growth-restricted newborns [14]. Thus, pregnancy is a critical period for programming children’s metabolism in a manner that influences their long-term risk [13,14]. In addition, during early infancy, a rapid weight gain may result in an increased risk for obesity and non-communicable diseases (NCDs) in childhood and beyond [15,16,17].

Leptin is also produced by the mammary gland but can pass to breast milk from maternal serum, and the significate of maternal leptin has been previously investigated with contrasting results [18,19]. In recent years, maternal milk leptin has been evaluated for its potential role in the postnatal programming of healthy phenotype in adulthood [2].

This study aims to assess the contribution of maternal plasma, cord blood (CB), and breastmilk leptin concentrations to neonates’ weight trajectory during the first year of life in different neonatal populations.

## 2. Methods

### 2.1. Study Design and Patients

This was a prospective longitudinal observational study. The Bioethical Committee of the Hospital approved the study and written informed consent was obtained from both parents before the neonate’s enrollment. The results described herein have never been reported before, nevertheless this cohort was part of a population previously evaluated for cord blood and maternal and neonatal leptin from birth to 3 months of age, from the same authors [20]. This study pertains to fully breastfed infants only, evaluated during the first year of age. The inclusion criterion was breastfeeding until the end of the 3 months of life. We measured maternal plasma leptin one week before delivery, umbilical CB at birth and mature breast milk (one sample in the first two weeks from the delivery). Maternal anthropometric and clinical data and blood samples were collected before delivery; children’s data were collected at birth and after 1 year of life.

Forty-five mother-infant dyads exclusively breastfeeding until the end of the third month of life could be enrolled. Dyads consisted of 16 full-term (FT), 16 preterm (PT), and 13 intra-uterine growth-restricted newborns (IUGR) and their mothers. FT babies included healthy newborns, gestational age (GA) ≥ 37 weeks, with adequate for gestational age (AGA) bodyweight. Infants born at GA 36 + 6 weeks of gestation or below were included in the PT group; IUGR newborns were born to mothers with altered flussimetric findings during ultrasound imaging and did not reach their genetically determined potential growth.

Exclusion criteria were infants with congenital malformations or infection, genetic syndromes, formula-fed, and infants large for gestational age.

### 2.2. Procedures

Leptin concentrations in maternal plasma and CB were determined as reported in the previous study [20]. Briefly, all plasma samples were processed simultaneously using Simple Plex™ Assays running on the Ella™ Automated immunoassay analyzer (ProteinSimple, Biotechne brand powered by R&D Systems, Inc., Minneapolis, MN, USA). (https://www.rndsystems.com/products/simple-plex-human-leptin-cartridge_spckb-ps-000309, last accessed on 14 October 2018).

Maternal milk samples (30 mL) were obtained at the end of the first morning breastfeeding, by pumping manually for five minutes, and were stored at −80 °C until analyses.

To separate fat from the aqueous phase, all samples were thawed, vortexed, and centrifuged (4000× *g*) for 30 min at 4 °C, to separate fat from the aqueous phase. Leptin concentration in freshly skimmed milk samples were determined as for plasma samples. The Low limit of quantification (LLOQ) is 2.20 pg/mL; the Upper LOQ (ULOQ) is 13,810 pg/mL. The limit of detection is 1.71 pg/mL. The intra-assay precision (within an assay and each control was tested 16 times in one assay) is 93.2 ± 8.70 pg/mL for low QC and 4319 ± 143 for high quality control (QC). The inter-assay precision (between assays, replicates of each QC were tested in multiple assays performed by 3 technicians using 2 lots of reagent, QC was tested 16 times in one assay): 94.7 ± 9.70 pg/mL for low QC and 4305 ± 178 pg/mL for high QC.

### 2.3. Statistical Analysis

Quantitative variables were summarized as median and Interquartile range (IQR, 25th−75th percentiles). Qualitative variables were described as counts and percentage. The Kruskal–Wallis and Fisher exact tests were applied to compare quantitative and qualitative variables between the three groups of neonates (FT, PT and IUGR). For the analysis, we used a zero-skewness log-transformation of leptin concentrations. Generalized linear regression models, with Sandwich standard errors (to allow for intragroup correlation), were fitted to assess the association of the log-transformed leptin and the increase of offspring weight, while adjusting for neonate type and time. Using residuals derived from the model, the partial correlation of leptin and weight, was computed together with its 95% bootstrapped confidence interval (95% CI), overall and by neonate type. The corresponding scatterplots are presented. The interaction of leptin concentration and groups was tested to assess whether any effect modification by group was present.

A 2-sided *p*-value < 0.05 was considered statistically significant. The Bonferroni correction was used for post hoc comparisons. Data analysis was performed with the STATA statistical package (StataCorp. 2019. Stata Statistical Software: Release 17. StataCorp LLC: College Station, TX, USA).

### 2.4. Study Endpoints

To assess the association of leptin concentrations in maternal plasma, CB and maternal milk with growth trajectories up to one year of age in all the neonates, adjusting for confounders (time and neonate types).

To verify whether the associations depend on the specific subgroup of mother-neonate dyads: FT, PT, and IUGR.

## 3. Results

Maternal plasmatic leptin concentrations were highest in the PT group, followed by the IUGR and FT groups, without reaching statistical differences (*p* = 0.213). Conversely, CB leptin concentrations were highest in FT and lowest in PT and IUGR neonates (PT vs. FT and IUGR vs. FT *p* < 0.001, both). Maternal milk concentration showed large variability within the same group, without significant differences.

At 12 months of age, PT and IUGR change in weight was higher (absolute value), although not significantly, than FT neonates (*p* = 0.272), while percent (%) change in weight was statistically significant in PT compared to FT, similarly to IUGR infants compared to FT (*p* < 0.001, both) (Table 1, Appendix A Appendix A; Appendix A). 

### 3.1. Association of Leptin Concentrations and Neonate’s Weight Trajectories

Maternal plasma leptin concentration (log-transformed) was significantly associated with infants change in weight (*p* = 0.017), adjusting for time and neonate type (confounders) in the multivariable analysis; the partial correlation of maternal plasma leptin and change in weight was −0.27 (95% CI −0.48 to −0.05).

A marginally non-significant (*p* = 0.051) association of CB leptin concentration and weight change was observed. The partial correlation for confounders was −0.30 (95% CI −0.58 to 0.0).

Finally, maternal leptin milk concentration (log-transformed) was significantly associated with neonates’ weight change (*p* = 0.006). The leptin-change in partial weight correlation was −0.33 (95% CI −0.51 to −0.15).

### 3.2. Association of Leptin Concentrations and Neonate’s Subgroups Weight Trajectories

Figure 1 illustrates the association of change in weight and leptin for each compartment and each different type of neonates, together with the corresponding partial correlations. Neonatal characteristics did not significantly modify the relationship of weight changes and maternal plasma leptin concentrations in PT and IUGR compared to FT (*p* for interaction = 0.268); however, partial correlations ranged from −0.04 (FT) to −0.28 (PT) and to −0.52 (IUGR).

In the case of CB leptin concentration, the effect modification by neonate type reached statistical significance (*p* = 0.026) and a significant partial correlation was elicited in the PT group (partial R −0.59 for PT; R −0.02 for IUGR; R −0.05 for FT). No significant modification by neonate type was found when considering the residual analysis from the multivariable regression plotting maternal milk and growth trajectory (*p* for interaction = 0.120). The inverse correlation was heterogeneous between the neonate’s types and was highest in the PT group (partial R −0.50 for PT; R −0.27 for IUGR; R 0.04 for FT).

## 4. Discussion

We investigated the relationship of maternal plasma, cord blood, and milk leptin concentration with offspring’s growth trajectories up to 1 year of age in three different mother-neonate dyads, FT, PT, and IUGR.

The multivariable analysis, adjusted for confounders, showed an inverse though weak correlation between leptin measured in the three different biological samples and growth trajectories up to one year of age. Maternal plasma leptin concentration was associated with infants change in weight (*p* = 0.017) and confirmed in the partial correlation when adjusting for time and neonate type (−0.27). Previous studies reported no association between maternal leptin concentration and birthweight/weight change [6,20]; or higher incidence of IUGR neonate in pre-eclamptic pregnancy, if maternal serum leptin levels were elevated [21]. Moreover, a significant rate of low-birth-weight infants in non-diabetic mothers with high leptin plasma concentration was reported [22,23]. Our results highlighted a marginally non-significant association between CB leptin concentration and weight change at one year of age for all neonates.

Conversely, the analysis of the specific sub-group demonstrated that PT neonates elicited a significant negative correlation. Other studies investigated the relationship between CB leptin concentration, birthweight, and growth pattern. Fonseca et al. [24] demonstrated a negative association between CB leptin with BMI and length change during the first six months of life of preterm infants, furthermore lower leptin levels correlated with greater catch-up growth. Equally, Karakosta et al. [25] evaluated the association of CB leptin and growth trajectories up to four years. Children with higher leptin concentration resulted in lower height, weight and BMI until early childhood; at the same time, a negative association was observed in small for gestation age infants and children with rapid infant growth during the first three months of life. Conversely, Chaoimh et al. [8] showed that cord blood leptin was positively associated with fat mass at birth, while higher concentrations were associated with slower weight gain in early infancy.

Since the preterm delivery leads to a premature separation from the maternal/placental leptin source, the infant is predisposed to postnatal hormone deficiency, enhanced by the lack of adipose tissue, which increases during the last three months of fetal life. Thus, leptin in the breastmilk could be a substitute source for neonates, particularly those born preterm [26]. In our study, leptin concentrations in maternal milk resulted in a similar small amount in each group, lower than the corresponding levels in maternal plasma and CB. Anyhow, maternal milk leptin was negatively associated with all the neonates’ change in weight. In particular, the sub-group analysis showed the most significant values plotting mother’s milk with PT neonate’s growth trajectories.

Aydin et al. [27] confirmed that leptin levels in colostrum (2.01 ng/mL) and mature milk (2.04 ng/mL) were more than five-fold lower than the corresponding blood levels (11.54 ng/mL). Schuster et al. [12] measured leptin concentrations in maternal milk 22-fold lower than the maternal serum; the same study showed a negative association between breast milk leptin and the infant’s weight increase during the first six months of life.

Dundar et al. [28] reported that maternal milk from small, large and adequate for gestational age infants had different leptin levels during the first months of life. Indeed, small for gestational age infants had a significant milk leptin reduction associated with higher weight gain. Andreas et al. highlighted no significant associations between milk leptin concentration and the timing of sample collection (measured at one week and three months post-partum) and hind and foremilk, demonstrating a constant concentration of the hormone through the first 3 months of lactation [29]. On the other hand, Kratzsch et al. [18] showed a variability of breast milk leptin concentrations depending, in particular, by maternal adipose tissue. In our study, maternal BMI, before pregnancy and at the delivery, resulted homogenous, without statistical differences among the three groups of mothers. In parallel, leptin concentrations in maternal milk resulted as well alike in the three groups and statistically associated with neonates’ change in weight. Besides maternal influences, the role of breastmilk leptin concentration is still a matter of debate: a recent review highlighted contrasting findings concerning the potential action of milk leptin on modeling growth, either showing infants’ association to weight changes or reporting no correlation [30].

Moreover, a recent study on 50 preterm infants found an association between higher maternal leptin intake and higher weight gain at 36 weeks of post menstrual age, suggesting a potential role for breastmilk leptin in earlier neonatal weight increase [31], which contrast with the known “appetite-inhibiting” role of the hormone [2].

Our study demonstrated for all group an inverse association between leptin of breast milk and neonatal growing trajectories, which was stronger in newborns born preterm. The outline of change in weight in the first year of life was significantly associated with low hormone levels: small and critical newborns struggle to rescue the appropriate size over time. Low leptin concentration contributes reaching the ideal body weight, avoiding the satiety perception, increasing feeding need. We found the most significant correlation between leptin concentration in each biological sample and weight gain in newborns born preterm.

Leptin concentrations were demonstrated to play a role in defensive mechanisms for survival, enhancing hunger, and expansion of energy stores needed for rapid growth. On the other side, monitoring for abnormal “catch-up growth” is mandatory. The association between excessive growth in infancy and overweight/obesity, insulin resistance, and elevated blood pressure in childhood or adulthood has been reported [32]. Therefore, measuring growth trajectory and leptin concentration is a valid tool for monitoring children’s health. In particular, the period of early infancy known as “the first 1000 days of life”, is a narrow but essential window for promoting future wellbeing [33,34].

During pregnancy, prevention measures can be adopted to promote healthy fetal growth and programming [17]. Similarly, in the post-natal period, it is possible to avoid future obesity and metabolic diseases, as an example, with prolonged breastfeeding that acts on an infant’s microbiota and immune system. [35].

The low number of subjects included in our study requires caution in the interpretation of results, and we did not include other perinatal and postnatal factors that have a part in children growth. Contrariwise, the inverse correlation between leptin in PT infants and change in weight was significant throughout the first year of age, regardless of the source of the biological compartment.

We speculate that leptin plays a role in neonatal growth trajectories, characterized mainly by an inverse correlation with leptin concentration in maternal plasma and breastmilk. The PT infants elicited the highest negative association with each leptin source. Low hormone levels appeared to contribute to critical neonates’ ability to recover a correct bodyweight similar to full-term children at 1 year. An eventual non-physiological “catch-up growth” should be monitored, and leptin perinatal levels may be an indicative tool. Further investigations are needed to strengthen the results.

## Figures and Tables

**Figure 1 nutrients-14-01451-f001:**
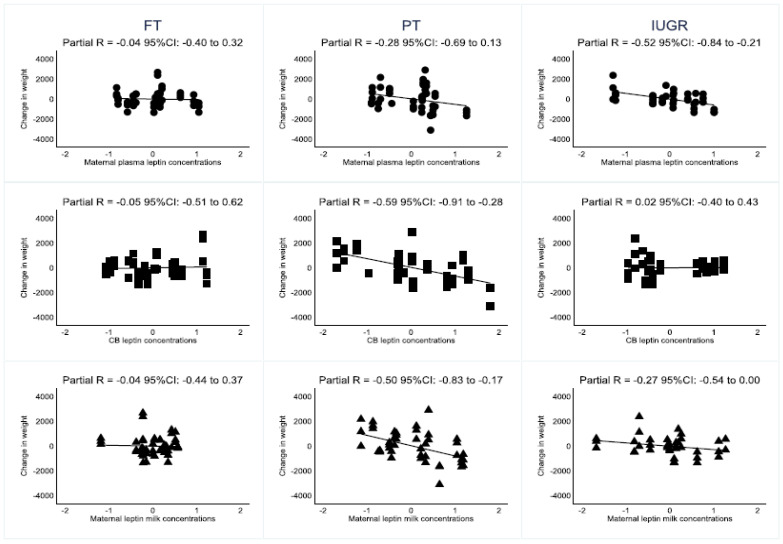
Partial correlations of leptin concentrations and change in weight by type of neonate. Residuals from the multivariable regression are plotted to account for time. Circles = maternal plasma leptin concentrations. Squares = CB leptin concentrations. Triangles = milk leptin concentrations.

**Table 1 nutrients-14-01451-t001:** Demographic data of mother-newborn couples, leptin concentrations, and weight increase in the three groups.

	FT (*n* = 16)	PT (*n* = 16)	IUGR (*n* = 13)	*p*-Value	Post Hoc Comparison *p*-Value—Bonferroni
Maternal age, years ^#^	31 (30–32)	32 (29–36)	32 (30–37)	0.377	-
Cesarean section delivery ^§^	1 (6.3%)	11 (68.8%)	10 (76.9%)	<0.001	PT vs. FT: 0.002
					IUGR vs. FT: <0.001
					IUGR vs. PT: >0.90
Maternal BMI before pregnancy *	23.5 (3.65) (range 16.5–29.8)	23.5 (3.98) (range 18.7–30.48)	23 (3.68) (range 19.0–31.2)	0.814	-
Maternal BMI at delivery *	28 (3.63) (range 20.8–33.2)	27.5 (4.0) (range 20.8–35.0)	26.0 (3.35) (range 220–33.6)	0.228	-
Male neonate ^§^	8 (50%)	7 (43.8%)	5 (38.5%)	0.929	-
Gestational Age, weeks ^#^	40.5 (39.4–41.0)	33.7 (29.7–34.6)	36.6 (32.1–37.3)	<0.001	PT vs. FT: <0.001
					IUGR vs. FT: <0.001
					IUGR vs. PT: 0.217
APGAR 1′	10 (9–10)	7.5 (5.5–9)	9 (6–9)	<0.001	PT vs. FT: <0.001
					IUGR vs. FT: 0.027
					IUGR vs. PT: 0.304
APGAR 5′	10 (10–10)	9 (8–10)	10 (7–10)	0.022	PT vs. FT: 0.003
					IUGR vs. FT: 0.100
					IUGR vs. PT: 0.402
Maternal plasmatic leptin concentration pg/mL ^#^	44,473.5 (22,214.5–50,302.4)	75,575.2 (27,219.9–89,243.3)	71,768.5 (41,890.1–12,1378.1)	0.213	-
Maternal milk leptin concentration pg/mL ^#^	620.6 (462.3–881.6)	622.0 (329.7–1235.0)	844.3 (444.9–1008.9)	0.782	-
CB leptin concentration pg/mL ^#^	19,280.5 (11,800.8–32,894.8)	3958.1 (1982.9–11,545.6)	1588.8 (1223.1–6912.0)	<0.001	PT vs. FT: <0.001
					IUGR vs. FT: <0.001
					IUGR vs. PT: 0.632
Birth weight g ^#^	3242.5 (3022.5–3585.0)	1802.5 (1367.5–2203.5)	1770.0 (1265.0–2265.0)	<0.001	PT vs. FT: <0.001
					IUGR vs. FT: <0.001
					IUGR vs. PT: >0.90
Change in weight g ^#^ (increase at 12 months)	6135 (5893–7110)	7062 (6275–8565)	6295 (5970–6955)	0.272	
% Change in weight (increase at 12 months)	191.7 (183.4–224.8)	374.9 (288.9–564.6)	341.6 (288.5–549.8)	<0.001	PT vs. FT: <0.001
					IUGR vs. FT: <0.001
					IUGR vs. PT: >0.90

* mean and (SD); ^§^ n and (%); ^#^ median and (IQR 25th–75th percentiles). Abbreviations, FT: full-term; PT: preterm; IUGR: intrauterine growth restriction; BMI: body mass index.

## Data Availability

Not applicable.

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
