# Peer review of "Leptin Levels of the Perinatal Period Shape Offspring’s Weight Trajectories through the First Year of Age"

_nutrients, 2022, doi:10.3390/nu14071451_

Round 1
Reviewer 1 Report
I appreciate the opportunity to review this paper.
Authors present the weight change after 1 year by % weight (increase at 12 months) – That’s important to understand the weight gain, but when comparing term with PT and IUGR infants, it’s also important include a variable that show the “final result”, this means how those infants accomplish in the recovering of weight after one year, I suggest present results also by weight percentile at 1 year. This will help to understand if PT and IUGR achieve a “normal” weight after 1 year, and better understand the link with leptin values
In discussion, authors say leptin levels in maternal milk are lower than the corresponding levels in maternal plasma and CB. Authors should point the high variability of leptin in human milk, since they are dependent on a plenty of influencing factors with an important relevance of maternal anthropometric characteristics (Kratzsch J, et al. Best Pract Res Clin Endocrinol Metab. 2018 Jan;32(1):27-38.)
Table 1 – put units of Birth weight and weight gain
Suggest use the same terminology in all tables/fig - Weight (increase at 12 months) (table 1) OR change in weight (fig 1)
Writing: review – abstract “CB), mature breast milk and correlating “; missing coma on pag 6 “lower height, weight and…” and lack of point in the end of reference 11; pag 6 – instead of use abbreviation SGA, use “small for gestational age”; pag 3 in Procedures –put “quality control (QC)”, primary to “QC”
Reviewer 2 Report
In this study authors aimed to assess the contribution of maternal plasma, cord blood (CB) and
breastmilk leptin concentrations to neonates’ weight trajectory during the first year of life
in different neonatal populations.
Today several studies have demonstrated a significant role of normal leptin levels for fetal and neonatal growth and development, and most of them support that normal leptin production during pregnancy is a factor responsible for normal gestation, embryonic/fetal development, and fetal growth. Studies have been conducted and involved concurrent leptin measurements in maternal serum at delivery, in cord blood in mother-infant pairs as well in breast milk.
Authors in this study concluded that” We conclude that leptin might play a role in neonatal growth trajectories, characterized mainly by an inverse correlation with leptin concentration in maternal plasma and breastmilk. The PT infants elicited the highest negative association with each leptin source. Low hormone levels contributed to neonates’ ability to recover a correct bodyweight similar to full-term children at 1 year. An eventual non-physiological “catch-up growth” should be monitored, and leptin perinatal levels may be an indicative tool.” It is very strong conclusion and based in the fact that many perinatal and postnatal factors such as (BMI of mothers or nutritional practices, primarily breastfeeding, and child illness) are likely to influence growth during infancy and no adjusting was done for this cofounders. l believe this is an important limitation of the study which weakens the study result and conclusions.
Author Response
Please see the attchment

Reviewer 3 Report
In this study the authors tried to evaluate the impact of maternal plasma, cord blood (CB) and breastmilk leptin concentrations to neonates’ weight trajectory during the first year of life in different neonatal populations.
There are too many factors determining an infant’ s growth till the 12th month of age but the authors have not analyzed these confounders. Perhaps it would be more correct to measure the concentration of leptin plasma levels in the infant’ s blood at this age or to perform serial measurements in mother’s breast milk for the first three months period.
It is well known the correlation between mother’s leptin plasma levels and her BMI, but the authors have not performed adjustment for this confounder.
In results section authors noted that “Maternal plasmatic leptin concentrations were highest in the PT group, followed by the IUGR and FT groups, without reaching statistical differences (p=0.213). The percent (%) weight increase at 12 months was statistically significant in PT compared to FT, similarly to IUGR infants compared to FT (p<0.001, both), PT and IUGR weight gain was higher, although not significant, than FT neonates”. Contrarily, in discussion section the authors mention that “specifically, maternal leptin plasma concentration was statistically inversely associated with all the neonates' weight change and confirmed this correlation when considering each subgroup individually.”
Round 2
Reviewer 2 Report
no additional comments or concerns
Reviewer 3 Report
I thank the authors for the clarified anwers to my comments.
There is only a grammatical mistake in the last line of the abstract in the revised version that the authors should correct: "further investigation is needed"...